# Impact of the first year of COVID-19 vaccination strategy in Brazil: an ecological study

Soraida Aguilar ®,[1] Leonardo S L Bastos ®,[1,2] Paula Maçaira ®,[1] Fernanda Baião,[1] Paulo Simões,[1] José Cerbino-Neto ®,[3,4] Otavio Ranzani,[5,6] Silvio Hamacher,[1,2] Fernando A Bozza ®[3,4,7]

For numbered affiliations see end of article.

**Correspondence to**
Dr Fernando A Bozza;
bozza.fernando@gmail.com

## ABSTRACT

**Objectives** No consensus exists about the best COVID-19 vaccination strategy to be adopted by low-income and middle-income countries. Brazil adopted an age-based calendar strategy to reduce mortality and the burden on the healthcare system. This study evaluates the impact of the vaccination campaign in Brazil on the progression of the reported COVID-19 deaths.

**Methods** This ecological study analyses the dynamic of vaccination coverage and COVID-19 deaths in hospitalised adults (≥20 years) during the first year of the COVID-19 vaccination roll-out (January to December 2021) using nationwide data (DATASUS). We stratified the adult population into 20–49, 50–59, 60–69 and 70+ years. The dynamic effect of the vaccination campaign on mortality rates was estimated by applying a negative binomial regression. The prevented and possible preventable deaths (observed deaths higher than expected) and potential years of life lost (PYLL) for each age group were obtained in a counterfactual analysis.

**Results** During the first year of COVID-19 vaccination, 266 153 517 doses were administered, achieving 91% first-dose coverage. A total of 380 594 deaths were reported, 154 091 (40%) in 70+ years and 136 804 (36%) from 50-59 or 20-49 years. The mortality rates of 70+ decreased by 52% (rate ratio [95% CI]: 0.48 [0.43-0.53]) in 6 months, whereas rates for 20–49 were still increasing due to low coverage (52%). The vaccination roll-out strategy prevented 59 618 deaths, 53 088 (89%) from those aged 70+ years. However, the strategy did not prevent 54 797 deaths, 85% from those under 60 years, being 26 344 (45%) only in 20–49, corresponding to 1 589 271 PYLL, being 1 080 104 PYLL (68%) from those aged 20–49 years.

**Conclusion** The adopted aged-based calendar vaccination strategy initially reduced mortality in the oldest but did not prevent the deaths of the youngest as effectively as compared with the older age group. Countries with a high burden, limited vaccine supply and young populations should consider other factors beyond the age to prioritise who should be vaccinated first.

## INTRODUCTION

The COVID-19 pandemic imposed unprecedented challenges worldwide. In 2020, the pandemic in Brazil quickly spread across the

## STRENGTHS AND LIMITATIONS OF THIS STUDY

⇒ We evaluated the impact of the COVID-19 vaccination strategy in Brazil on the number of reported deaths during the campaign's first year. In 2021, 266 153 517 COVID-19 vaccine doses were administered in adults with 91% vaccination coverage (first dose).

⇒ The vaccination strategy followed an age calendar and prioritised the high-risk groups, and following an age-based calendar, the mortality rates of the elderly population decreased earlier than those of the young population.

⇒ Alternative strategies to an age-based calendar are necessary for countries with low resources and young populations. These strategies should consider the local burden, vaccination supply and speed, vaccine hesitancy and socioeconomic vulnerability.

⇒ Non-pharmacological interventions that may affect the progression of deaths and mortality rates were not considered in the analysis.

⇒ This ecological study analysed notified data in the SIVEP-Gripe (*Sistema de Informação de Vigilância Epidemiológica da Gripe*, which limits the assessment of all COVID-19 deaths and excess mortality in the country.

different states, with distinct patterns and burdens due to the disparities in health assistance, income and a lack of a unified national strategy to face the pandemic.[1 2] With the emergence and dominance of SARS-CoV-2 Variants of Concern (VOC)[3] at the end of 2020,[4] characterised by high transmissibility,[4] immune escape[5 6] and increased disease severity, substantial pressure was imposed on the health system. At the beginning of 2021, when the vaccination campaign started in the country, a new surge—the second COVID-19 Gamma wave—was still rampant, with higher mortality than the first.[3]

Countries have used different strategies to define who vaccinates first.[7 8] Several studies evaluated the impact of vaccination campaigns on infections, hospitalisations

and saving lives principally in high-income countries.[8–11] In Brazil, the National Immunization Program (PNI), responsible for providing the vaccination guidelines, is part of the Brazilian universal health system (*Sistema Unico de Saúde—SUS*) and has been recognised as a model of vaccination campaigns and population adhesion worldwide.[12] The Brazilian COVID-19 vaccination campaign, which started on 17 January 2021, prioritised high-risk groups and considered an age-based calendar beginning with the elderly (85 years or more).[13] Vaccine distribution accomplished the national guidelines centralising the distribution of vaccine doses to the states and municipalities.

It is still unclear how to best prioritise groups for mass vaccination during pandemics/outbreaks such as COVID-19, especially in low- and middle-income countries (LMICs) with a high burden of the disease and a young population.[14–16] Hence, we performed an ecological study to evaluate the impact of the vaccination campaign in Brazil on the progression of the reported COVID-19 deaths. We estimated the number of prevented and possible preventable deaths during the campaign stratified by age groups.

## METHODS
### Study design and participants
We conducted an ecological study to analyse the progression of reported COVID-19 deaths and mortality rates before and during the first year of the vaccination campaign roll-out in Brazil.

Daily data on reported deaths were obtained from the Influenza Epidemiological Surveillance Information System (*Sistema de Informação de Vigilância Epidemiológica da Gripe*, SIVEP-Gripe), a nationwide surveillance open-access database used to monitor severe acute respiratory infections in Brazil. SIVEP-Gripe has been the primary source of reported COVID-19 deaths in the country. This database covers all Brazilian municipalities; however, in-hospital people belong to 80% of these municipalities, where those territories correspond to 96% of Brazil's population.[2] Each register contains information about the individual's demographics, self-reported symptoms, comorbidities, hospital and intensive care unit (ICU) admission and ventilatory support, in-hospital outcome (death or discharge) and dates of symptom onset, hospital admission and ICU admission. We included all adult participants (≥20 years old) with SARS-CoV-2 confirmed infection (Reverse transcription polymerase chain reaction [RT-PCR] or clinical-epidemiological criteria) whose death was reported between 26 February 2020 and 31 December 2021.

Daily data on COVID-19 vaccination coverage were obtained from the Brazilian National Immunization Program database (SI-PNI). The SI-PNI contains information on the individual's demographics (age, sex and place of residence), vaccine dose and platform and place and date of dose administration. We included all doses

(first, second or single) administered to adults (≥20 years old) between 17 January 2021 (the start of the vaccination campaign) and 31 December 2021. Finally, we used the 2020 population estimates provided by the Brazilian Ministry of Health stratified by age.

Data were publicly available, anonymised and de-identified. Following ethically agreed principles on open data, this analysis did not require ethical approval in Brazil. Information on the data sources is shown in online supplemental table 1.

### Patient and public involvement
No patient was involved.

### COVID-19 vaccination campaign roll-out in Brazil
On 17 January 2021, the Brazilian PNI initiated the vaccination campaign for COVID-19. Vaccine doses were freely distributed to states and municipalities, which managed the administration according to the availability of resources. The COVID-19 vaccination campaign strategy followed a national age-based calendar during the whole period, with some local adaptations. It started in parallel vaccinating those aged 90+, Indigenous individuals, healthcare workers and high-risk groups based on comorbidities and then followed an exclusively age-based calendar. Four vaccine platforms were considered in the country: two doses of Sinovac-CoronaVac, ChAdOx1-S/nCoV-19 (Oxford-AstraZeneca) or BNT162b2 (Pfizer/BioNTech), or Ad26.COV2.S (Janssen) single-dose.

### Outcomes
Our primary outcome was the estimated number of prevented COVID-19 deaths. Secondary outcomes comprise possible preventable deaths, the potential years of life lost (PYLL), the mortality rates per 100 000 population and the first-dose and second or single-dose vaccination coverage. All quantities, rates and estimates were evaluated temporally for the whole country and stratified by age groups; no further stratification was performed.

### Data analysis
We described continuous variables using medians and interquartile ranges (IQRs) or means and standard deviations (SDs) when applicable and frequencies and proportions for categorical variables. The analysis was prespecified, and the sample size was pragmatic (SIVEP-Gripe database). All analyses considered complete data, and no imputation of missing values was performed.

We evaluated the progression of COVID-19 deaths before and during the vaccination campaign (26 February 2020 to 31 December 2021) for the whole country and stratified by age groups. We generated the time series of daily reported deaths and mortality rates using the 15-day moving average to reduce the effects of variability in the notification process. Changes in the representativeness of age groups in the deaths were assessed by calculating the proportion of each group in the daily count. In addition, vaccination coverage of at least one dose ('partially vaccinated') and two or single doses ('fully vaccinated') was

calculated as the number of doses divided by the corresponding estimated population.

To estimate the impact of the vaccination campaign strategy on the evolution of daily reported mortality rates for each age group (20–49, 50–59, 60–69 and 70+), we performed a three-stage analysis.

First, we estimated the dynamic effects of the vaccination campaign to examine how age-specific death rates deviated from the national average before and after the onset of the vaccination campaign. The national average is represented by the national age-adjusted mortality COVID-19 rates in Brazil as a surrogate of the pandemic progression in the country during the vaccination campaign. We compared the daily mortality rates of each age group and the national age-adjusted mortality COVID-19 rates using rate ratios (RR). Afterwards, we applied a negative binomial regression model. The response variable ($y_t$) was the monthly COVID-19 reported deaths, and the covariates were the age group ($x_{age,t}$), the period indicator ($x_{period,t}$), separated into indicators, one for each month, their interaction ($x_{age,t} \times x_{period,t}$) and the population ($x_{population}$) as the offset variable. To obtain the dynamic effects for each age group, we used the national average as the reference. The effect was calculated as the exponentiated coefficient of the interaction terms, defined as the RR and their corresponding 95% confidence intervals (CIs). The negative binomial regression model is represented as follows:

$$\ln y_t = \begin{aligned} &\beta_0 + \beta_1 x_{age,t} + \beta_2 x_{period,t} + \beta_3 x_{age,t} \times \\ &x_{period,t} + offset\left(x_{population}\right) \end{aligned} \quad (1)$$

where $t$ represents the monthly observations and $\beta_0$, $\beta_1$, $\beta_2$ and $\beta_3$ are regression coefficients. Additionally, a sensitivity analysis is performed by exchanging the national average as a reference to the older and younger (70+ and 20–49) age groups.

Subsequently, we quantified the impact of the vaccination strategy in terms of prevented and possible preventable deaths for each age group using a counterfactual analysis.[17] Using a linear regression model, we predicted the mortality rate of an age group (response variable, $y_i$) based on the national age-adjusted mortality rates in the country (predictor variable, $x_i$).[17] The model is expressed by:

$$y_i = \beta_0 + \beta_1 x_i + \varepsilon_i \quad (2)$$

where $\varepsilon_i$ denotes the error term, $i$ is the daily observations of the target age group and $\beta_0$ and $\beta_1$ are regression coefficients. Although the national age-adjusted mortality rates in the country do not strictly represent a scenario without vaccination, we could obtain the number of expected deaths considering this progression as being the dynamic pandemic progression and the vaccination roll-out as the speed and effectiveness in the decrease of deaths for each age group.

We considered the start of the vaccination roll-out as the date of intervention onset (17 January 2021). Hence, we fit the model using data prior to the vaccination roll-out to obtain the predicted mortality rates for each age group in the following period. We multiplied the predicted mortality rates by the total population to obtain the number of expected deaths. When comparing the number of expected to observe deaths for each age group, we considered 'prevented deaths' as when the observed deaths were lower than the expected deaths; otherwise, we considered 'possible preventable deaths'.

Furthermore, we estimated the corresponding number of prevented and preventable PYLL for each age group, considering the midpoint of each age group and a life expectancy of 76 years for Brazil in 2019.[18] All analyses were performed in R 4.2.1. We followed Strengthening the Reporting of Observational Studies in Epidemiology (STROBE) recommendations.

### Role of the funding source

The funders had no role in any decision about the manuscript. All authors had full access to all the data in the study. SA, PM, FB and LSLB verified the data, and all authors approved the final version of the manuscript for publication.

## RESULTS

### Age-adjusted mortality and vaccine coverage in the first year of vaccine roll-out

From 26 February 2020 to 31 December 2021, 565 774 COVID-19 deaths were reported (figure 1A). Most of these comprised those aged 70 years or older (253 092, 45%), and one-third were those aged under 60 years (figure 1B). The daily average mortality rates per 100 000 population were the highest between March and May 2021 for all most of the age groups and occurred during the dominance of VOC Gamma (online supplemental figure 1A and online supplemental table 2). When comparing the proportion of reported deaths (figure 1C), the 70+ age group showed a decrease since March 2021, 2 months after the vaccination roll-out start, being the lowest in May to June 2021, almost 50% lower compared with the pre-vaccination period. During this period, the proportion of deaths related to 20–49 and 50–59 years increased. However, in late 2021, there was a gradual reduction in the proportion of deaths related to those under 60 years.

With respect to the vaccination roll-out, 266 153 517 vaccine doses for COVID-19 were administered in adults (≥20 years old) in 2021, with 137 644 737 (51.66%) first doses and 128 508 780 (48.34%) second or single doses. At the end of 2021, the national first-dose vaccination coverage for adults was 90.7%, with 86.74% for 20–49 years, 94.75% for 50–59 years, 100.78% for 60–69 years and 99.59% for 70+ years (online supplemental tables 3 and 4). The progression of vaccination coverage rates was different for each age group due to the campaign strategy (figure 1D). Brazil achieved 75% vaccination coverage in August 2021, 6 months after the vaccination roll-out. In the same period, the 70+ group had 98% first-dose coverage, whereas the 20–49 group presented 52% coverage (online supplemental table 3). We observed

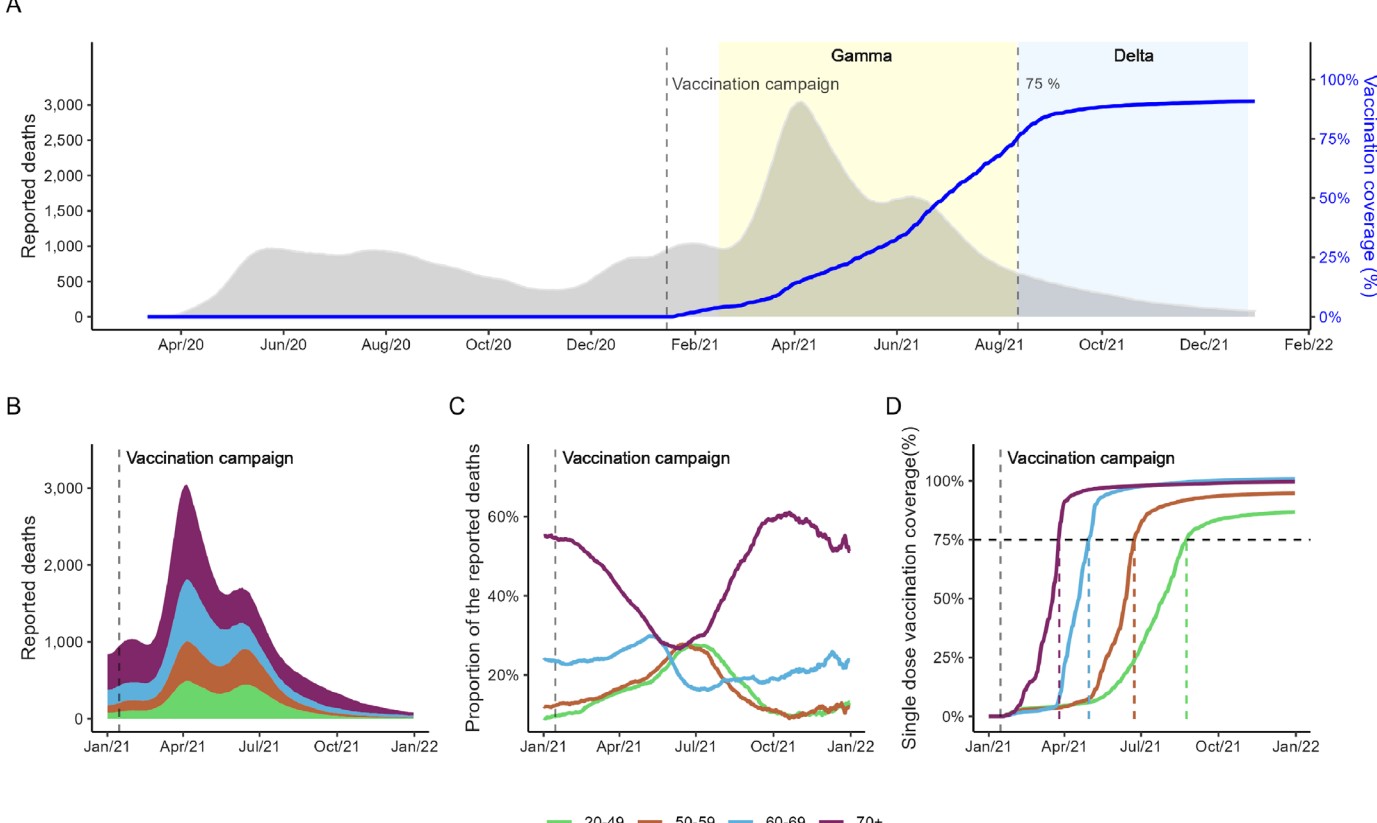

**Figure 1** Progression of COVID-19 reported deaths and the vaccination roll-out in Brazil: (A) daily number of reported deaths (15-day moving average, left y-axis) and first dose vaccine coverage (right y-axis) from 26 February 2020 to December 2021; yellow shaded areas display the period of Variants of Concern dominance (50% of samples using oubreak.info/GISAID data) for Gamma (15 February 2021) and Delta (11 August 2021)—vertical dashed lines indicate the start of the vaccination roll-out in Brazil and the 75% first vaccination coverage of adult population, respectively; (B) daily number reported deaths stratified by age group in 2021; (C) daily proportions of deaths reported for each age group (number of deaths/total number of deaths per day); (D) first dose vaccination coverage stratified by age group—vertical dashed lines refer to the start of the vaccination campaign for each age group, the horizontal dashed line indicates the 75% vaccination coverage target by the WHO.

that the oldest group (70+) already had more than 90% vaccination coverage by the end of March 2021. Similar behaviour for the second or single dose was observed (online supplemental figure 2 and online supplemental table S4).

### Vaccination effect on mortality rates

To assess the impact of the vaccination campaign strategy on reducing mortality rates, we compared the mortality rates for each age group with the national age-adjusted mortality rates in the country (figure 2A and online supplemental figure 1A). The 70+ group showed the highest mortality rates in the whole period (RR>1). However, 6 months later the vaccination roll-out started, the mortality rates reduced from 3.84 deaths per 100 000 population (RR: 6.16) in January 2021 to 3.31 deaths per 100 000 (RR: 3.11) in June 2021 (online supplemental table 2). In the same period, the 20–49 age group showed an increase in mortality rates (RR: 0.21 in January 2021 and 0.58 in June 2021) compared with the national age-adjusted mortality rates. After September 2021, the RR for all age groups returned to similar levels to those seen before vaccination (online supplemental table 2).

We estimated the dynamic effect of age-specific death rates and how they deviated from the national average using a negative binomial regression model to evaluate the association between the reported deaths, age group and the period indicator (figure 2B). When analysing the temporal progression of the RR stratified by age group (online supplemental table 5), the 70+ age group had the lowest rates in June 2021 (RR [95% CI]: 0.48 [0.43 to 0.53]), which evidenced a 52% decrease in the number of deaths in 5 months. With regard to age groups 50–59 and 60–69, we estimated a decrease in death rates of 15% and 30% (RR [95% CI]: 0.85 [0.75 to 0.96] and RR [95% CI]: 0.70 [0.63 to 0.71], respectively) for at least 4 and 6 months in the first year of vaccination (online supplemental table 5).

When performing the sensitive analysis (online supplemental figure 3 and online supplemental tables S6 and S7), the extreme age groups (20–49 and 70+) display a similar behaviour from those age groups obtained from the dynamic effect when comparing to the national average (figure 2B). Furthermore, evaluating the before and after the vaccination campaign (reference December/20) was

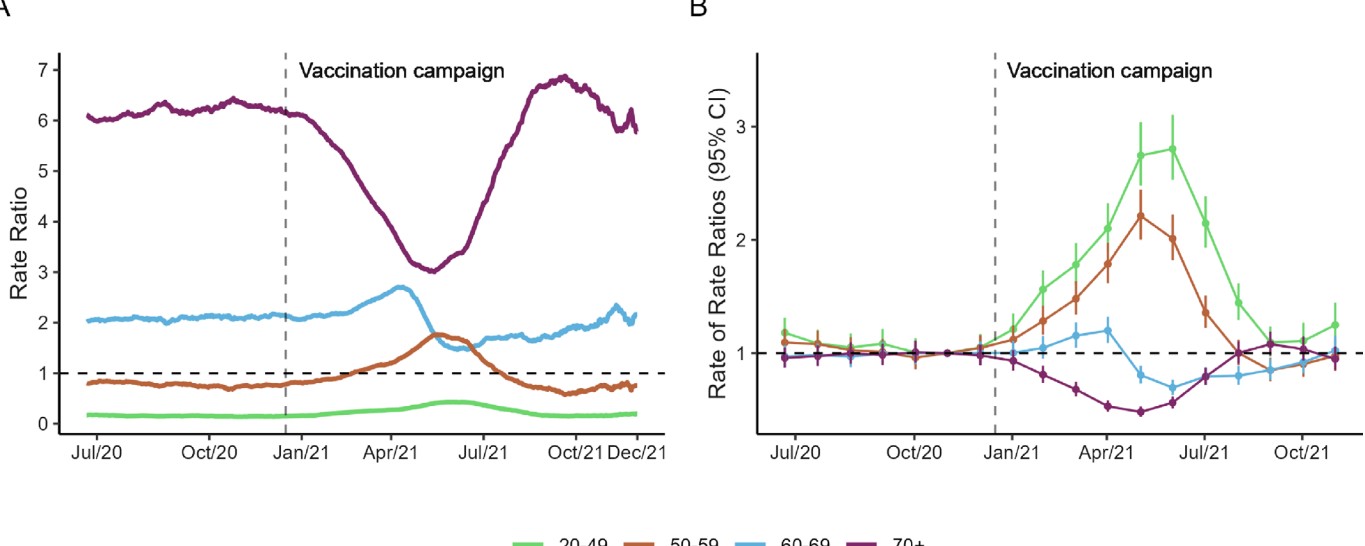

**Figure 2** The impact of the COVID-19 vaccination campaign in reported deaths stratified by age groups: (A) rate ratios comparing daily mortality rates stratified by age group with the national mortality rate; (B) estimated effect of the vaccination campaign in each age group. Effects were obtained as the rate ratio and their respective 95% CIs. The vertical dashed line refers to the start of the vaccination campaign on 17 January 2021. Our analysis considered 6 months before the vaccination roll-out as the baseline period and used the national mortality rates as a reference in a negative binomial regression model.

evidenced a rise on the mortality rates (online supplemental figure 1B and online supplemental table S8) after the start of the vaccination campaign which became increasingly higher for younger age groups (20–49 and 50–59). In June there was a second increase that was not repeated for the older age groups (60–69 and 70+).

### Number of prevented and possible preventable deaths and potential years of life lost

We quantified the number of prevented and possible preventable deaths in each age group during the vaccination roll-out (figure 3). We estimated that vaccination prevented 59 618 deaths compared with the expected deaths in the first year of vaccination roll-out (table 1). Of these, 89% were prevented in those aged 70 years or more (53 088), followed by those aged 60–69 years (6032, 10%), which achieved high vaccination coverage earlier than younger age groups (figure 1D).

We considered possible preventable deaths when the number of observed deaths was higher than those expected if each age group followed the progression of the national age-adjusted mortality rates in the country (figure 3, table 1). Despite the considerable number of deaths prevented, we estimated 54 797 potentially preventable deaths in the same period. More than 85% of possible preventable deaths comprised those under 60 years old, half of them corresponding to those aged 20–49 years (26 344, 48%), which achieved late vaccination coverage compared with older groups (figure 1D).

In addition, we estimated the impact of the vaccination roll-out in terms of the PYLL. The vaccination strategy prevented 371 894 PYLL, especially for those aged 60 years or older. However, it was not able to prevent 1 589

271 PYLL, composed mainly of those aged 20–49 years (68%) (table 1).

### DISCUSSION

We evaluated the impact of the COVID-19 vaccination strategy in Brazil on the number of reported deaths during the campaign's first year by analysing the progression of vaccination coverage and reported deaths and model the prevented and possible preventable deaths. In prioritising the vaccination of high-risk groups, the mortality rates of the elderly population reduced earlier compared with the young population, especially in a period of high burden due to the introduction of new VOCs. After 6 months of vaccination roll-out, the vaccination coverage for those aged 70+ years was 98% compared with only 52% for the age group 20–49, which resulted in a reduction of 44% and the mortality rates in the 70+ years and for those aged 20–49 were still increasing due to the low coverage. At the end of 2021, the first dose coverage achieved 91%, and the estimated number of prevented deaths was 59 618. However, the delay in the vaccination of younger ages may have resulted in an expressive number of deaths that could be prevented in this group.

Brazil was one of the epicentres of COVID-19 deaths in 2020, which intensified in a second and larger surge at the beginning of 2021 when the COVID-19 vaccination campaign started.[3] The first months comprised limited vaccine dose availability until May 2021. The strategy prioritised the immunisation of high-risk groups (ie, Indigenous, healthcare workers and comorbidities) and older ages, intending to reduce the burden of deaths and overload over the health system. This strategy resulted in

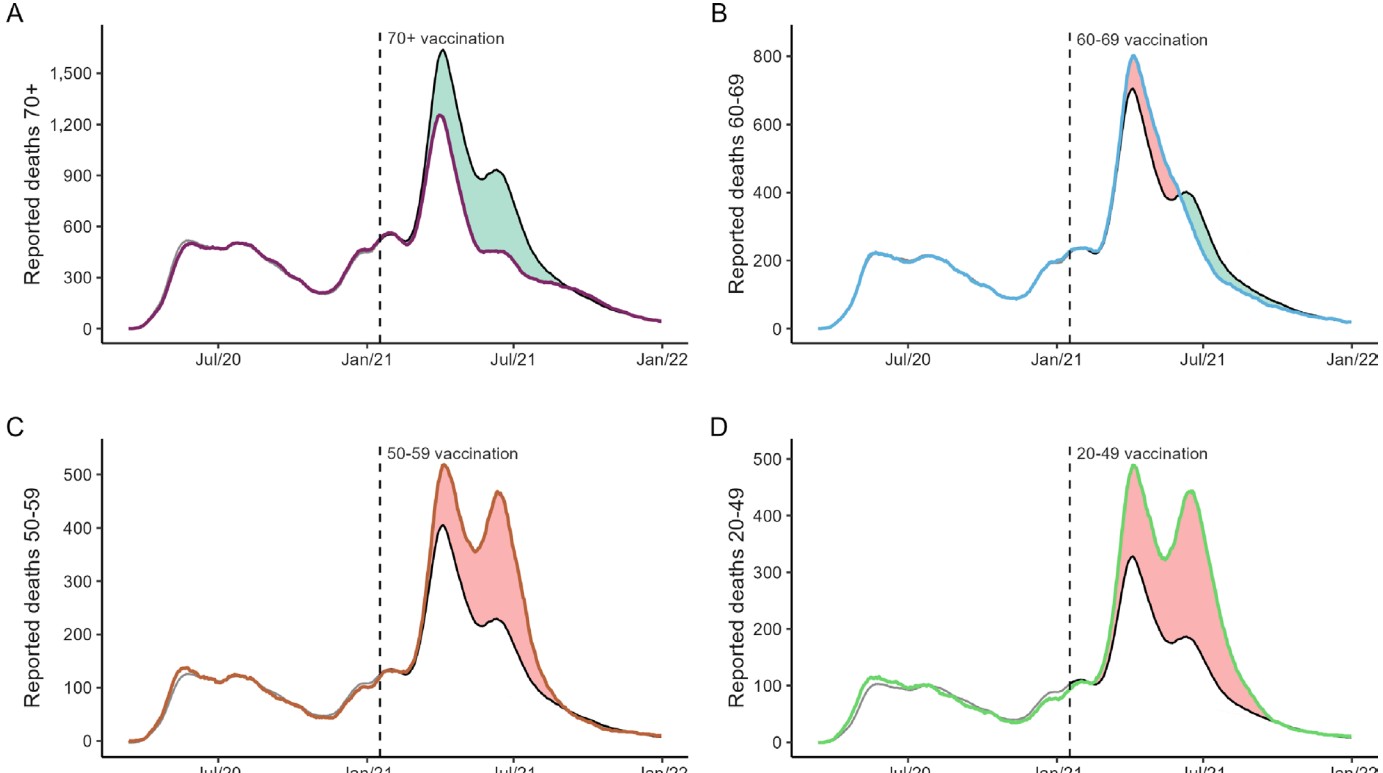

**Figure 3** Estimated prevented and possible preventable deaths stratified by age group. For each age group (coloured lines), the predicted deaths (black lines) were obtained whether that age group followed the national age-adjusted mortality rates. Grey lines correspond to the fitted linear model. Predicted deaths were estimated from the period when 10% of first-dose vaccination coverage was achieved for each age group. Prevented deaths (green-shaded area) were obtained if observed deaths were lower than predicted, and possible preventable deaths (red-shaded area), if the number of observed deaths was higher than predicted. The vertical dashed lines refer to the start of the vaccination campaign on 17 January 2021.

a rapid decrease in mortality rates in the country, driven mainly by those aged 70+, which displayed the highest number of prevented deaths (53 088) in the period. The strategy successfully prevented deaths in the older population; however, the major drawback was the expressive number of deaths in those aged <60 years old. Although non-pharmacological interventions (NPIs) such as social distancing and masks mandates have been implemented

in Brazil, there was not a coordinated and unified guideline at the national level. Due to the country heterogeneity, municipalities applied NPIs under different flags, but not with a high degree of adherence; despite this when the vaccination campaign began none NPIs continued.

Except for those with high-risk conditions, vaccination for younger groups started only 4–6 months after vaccination for older groups, during the third surge of deaths

**Table 1** Comparison between observed and predicted deaths during the vaccination campaign in Brazil in 2021, stratified by age group

| | Age group | | | | |
|---|---|---|---|---|---|
| **Results** | **20–49** | **50–59** | **60–69** | **70+** | **Total** |
| Observed deaths | 65 582 | 67 991 | 86 126 | 145 812 | 365 511 |
| Expected deaths | 39 393 | 48 098 | 85 064 | 197 777 | 370 332 |
| Prevented deaths | 155 | 343 | 6032 | 53 088 | 59 618 |
| Possible preventable deaths | 26 344 | 20 236 | 7094 | 1123 | 54 797 |
| Prevented PYLL*† | 6355 | 7203 | 66 352 | 291 984 | 371 894 |
| Preventable PYLL*† | 1 080 104 | 424 956 | 78 034 | 6177 | 1 589 271 |

Prevented deaths were obtained if observed deaths were lower than predicted, and possible preventable deaths, were obtained if the number of observed deaths was higher than predicted.
*PYLL was calculated using the average age for each group and a life expectancy of 76 years for Brazil in 2020.
†The denominator is the adult population of Brazil for this age group=151 778 738.
PYLL, potential years of life lost.

(July 2021). Although mortality risks for this population are lower compared with the elderly, Brazil is composed mainly of young people. At the end of 2021, 86 834 deaths were reported for the 20–49 age group, where 66 692 (77%) occurred during the vaccination roll-out period. The trend of mortality for the younger started to decrease in July 2021 and was more pronounced in September 2021, when vaccination coverage of this population was higher than 75%. Our analysis did not discriminate between prioritised high-risk groups from those individuals that following the age-based calendar in the vaccination campaign, since this is a low percentage of doses from these groups, especially those based on comorbidities lower than 10%, hence not significantly impacting the results or the progression of the vaccination campaign.

We estimated that the vaccination strategy prevented 59 618 deaths, composed mainly of those aged 60–69 or 70+ years. However, 54 797 possible preventable deaths of those aged 50–59 or 20–49 years occurred, corresponding to a potential savings of 1 589 271 PYLL, 68% from 20 to 49 years. These results align with previous studies that identified a reduction in deaths and hospitalisations due to vaccination in high-income countries.[8–11] A recent Brazilian study obtained 303 129 prevented deaths in the period from 17 January 2021 to 31 January 2022, with 61% from those aged ≥60 and 7% for the 20–39 age group.[19] Our estimates were obtained using a counterfactual analysis in which the control group represented the progression of national age-adjusted mortality rates during the vaccination campaign. Although it does not represent a scenario strictly without vaccination, we could obtain the number of expected deaths considering the dynamics in pandemic progression and the vaccination roll-out as the speed and effectiveness in the decrease of deaths for each age group. Hence, we were able to evaluate the results from the age-based calendar strategy and quantify the number of prevented and possible preventable deaths for each age group during the first year of the vaccination campaign.

Latin America was highly affected by the pandemic, the continent with the highest number of deaths per capita, with limited access to vaccine doses.[20 21] In Brazil, the availability of vaccine doses at the beginning of the vaccination roll-out was low due to the national government's late negotiation for vaccine purchases, global demand and the high concentration of doses in rich countries.[13 22] The concern with respect to vaccine hesitancy, encouraged by the country's leadership and polarised political environment, was not confirmed by previous evidence and high vaccination coverage was achieved.[13 22–24] The urgency of vaccination also refers to better use of a common resource, including access to vaccines, prioritising care for those most affected, that is, where the epidemic is behaving most intensely.[25]

Recently, the WHO revised the initial recommendation of prioritising the years of life saved to reduce deaths and disease burden, especially in scenarios of limited access to vaccines.[25] In Brazil, due to the low availability of vaccine doses at the start of the roll-out,

a strategy that followed an age-based calendar also prioritising high-risk groups was adopted. Our results demonstrated that the strategy protected the oldest. However, it could not significantly prevent the deaths of the youngest. These findings raise concerns about whether the age-based calendar is the best strategy for countries with low-resource settings and a young population.[26] Should alternative strategies prioritise locals with high burden and increase the years of life saved in locations with limited healthcare resources and socioeconomic vulnerability? Alternative strategies to the age-based calendar have been proposed pointing out the importance of considering the burden, age distribution, vaccination supply and speed, hesitancy and socioeconomic vulnerability to promote a better and more equitable decision-making process.[14 15 25 27 28] Other model approaches, such as agent-based or deterministic compartmental models, may also assess the potential gains of alternative strategies.[15 16]

The findings of this study should be interpreted in light of some potential limitations. First, we analysed data notified in the SIVEP-Gripe, which consists of cases reported by the surveillance system. This limits the assessment of all COVID-19 deaths and excess mortality in the country. However, the SIVEP-Gripe has been one of the country's central repositories of COVID-19 epidemiological data, with high information coverage.[2] Second, stringent policies and actions, as well as personal behaviour risk, may affect the progression of deaths and mortality rates and were not considered in the analysis. Third, although the vaccination roll-out started at the same time across the country, regional differences can indeed impact the vaccine coverage speed due to limited access or the age composition of certain locations.[13] Fourth, we did not take into account for previous infection, once the elderly had the highest rate of infection early in the pandemic, the natural immunity in this group might have also contributed to the same protection observed against death. Furthermore, the study has strengths: this is a nationwide data analysis of a large LMIC with a high burden of COVID-19; we estimate the dynamic impact of the vaccination programme in four age groups considering an average behaviour, which allows the assessment and understanding of the vaccination roll-out within the country.

## CONCLUSION

Our findings indicate that the vaccination campaign was able to prevent more than 59 000 deaths in Brazil, especially for the older population. However, the delay in the vaccination of younger ages may have resulted in an expressive number of deaths that could be prevented in this group.

**Author affiliations**
[1]Department of Industrial Engineering, Pontifical Catholic University of Rio de Janeiro, Rio de Janeiro, Brazil

²Tecgraf Institute, Pontifical Catholic University of Rio de Janeiro (PUC-Rio), Rio de Janeiro, Brazil
³Evandro Chagas National Institute of Infectious Diseases, Oswaldo Cruz Foundation, Rio de Janeiro, Brazil
⁴D'Or Institute for Research and Education, Rio de Janeiro, Rio de Janeiro, Brazil
⁵Barcelona Institute for Global Health, ISGlobal, Hospital Clínic-Universitat de Barcelona, Barcelona, Spain
⁶Pulmonary Division, Heart Institute (InCor), Hospital das Clinicas HCFMUSP, Faculdade de Medicina da Universidade de São Paulo, São Paulo, Brazil
⁷Comprehensive Health Research Centre (CHRC), NOVA Medical School, Universidade NOVA de Lisboa, Lisbon, Portugal

**Acknowledgements** We are grateful to all healthcare workers for their endless efforts to tackle the COVID-19 pandemic in Brazil. We thank the NOIS research group and the ICODA driver project EFFECT-Brazil team members for the discussions and collaborative production during the COVID-19 pandemic.

**Contributors** FB is the guarantor. SA, LSLB, PM, FB, PS, JC-N, OR, SH and FAB participated in the design and concept of the study. SA, LSLB and PM did the data curation. SA, LSBL, PM, FB, OR and FAB designed the data analysis. SA, LSLB, PM, FB, PS, SH and FAB performed the data analysis. SA, LSLB, PM, FB and FAB wrote the first version of the manuscript. LSLB, PM, FB, SH and FAB supervised the study. All authors had full access to data, participated in data interpretation, revised the manuscript and approved the final version of the manuscript.

**Funding** This work is part of the Grand Challenges ICODA pilot initiative, delivered by Health Data Research UK and funded by the Bill & Melinda Gates Foundation and the Minderoo Foundation. This study was also supported by the National Council for Scientific and Technological Development (CNPq), the Coordination for the Improvement of Higher Education Personnel (CAPES) - Finance Code 001, Carlos Chagas Filho Foundation for Research Support of the State of Rio de Janeiro (FAPERJ), the Pontifical Catholic University of Rio de Janeiro. OR is funded by a Sara Borrell grant from the Instituto de Salud Carlos III (CD19/00110). PM acknowledges suppor from the CNPq (Grant 311519/2022-9). OR acknowledges support from the Spanish Ministry of Science and Innovation and State Research Agency through the 'Centro de Excelencia Severo Ochoa 2019-2023' programme (CEX2018-000806-S) and support from the Generalitat de Catalunya through the CERCA programme. All authors carried out the research independently of the funding bodies. The findings and conclusions in this manuscript reflect the opinions of the authors alone.

**Competing interests** SH and FAB are funded by the CNPq and FAPERJ. PM is funded by CNPq (422470/2021-0) and FAPERJ (E-26/211.645/2021 and E-26/201.348/2022). OR is funded by a Sara Borrell fellowship from the Instituto de Salud Carlos III (CD19/00110), acknowledges support from the Spanish Ministry of Science and Innovation and State Research Agency through the 'Centro de Excelencia Severo Ochoa 2019–2023' programme (CEX2018-000806-S), support from the Generalitat de Catalunya through the CERCA programme and received a research grant from the Health Effects Institute for research unrelated to this manuscript. OR was also a member of the Data Safety Monitoring Board in the REVOLUTION and STOP-COVID trials, testing treatments against COVID-19, and is currently a member of the Data Safety Monitoring Board of the RENOVATE trial, testing respiratory support strategies in patients with acute respiratory hypoxaemic failure.

**Patient and public involvement** Patients and/or the public were not involved in the design, or conduct, or reporting, or dissemination plans of this research.

**Patient consent for publication** Not applicable.

**Ethics approval** Not applicable.

**Provenance and peer review** Not commissioned; externally peer reviewed.

**Data availability statement** All data relevant to the study are included in the article or uploaded as supplementary information. All data, including individual participant data, is publicly available with de-identification and anonymisation of patients. The data sources are described in the manuscript and in the supplementary material (online supplemental table S1). The raw data and code used for the analysis are available in a GitHub repository, with publication (https://github.com/noispuc/ICODA_COVID_VaccineStrategy).

**ORCID iDs**
Soraida Aguilar http://orcid.org/0000-0002-5177-8336
Leonardo S L Bastos http://orcid.org/0000-0001-7833-0403
Paula Maçaira http://orcid.org/0000-0001-6235-8006
José Cerbino-Neto http://orcid.org/0000-0001-9254-917X
Fernando A Bozza http://orcid.org/0000-0003-4878-0256

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
