## [Reviewer comments · BMJ Open]

ARTICLE DETAILS

TITLE (PROVISIONAL)	The impact of the first year of COVID-19 vaccination strategy in Brazil: An ecological study
AUTHORS	Aguilar, Soraida; Bastos, Leonardo S. L.; Maçaira, Paula; Baião, Fernanda; Simões, Paulo; Cerbino-Neto, José; Ranzani, Otavio; Hamacher, Silvio; Bozza, Fernando

VERSION 1 – REVIEW

REVIEWER	Mrejen, Matías Institute for Health Policy Studies
REVIEW RETURNED	03-Apr-2023

GENERAL COMMENTS	Comments for editor and author Thank you for the opportunity to review this paper on the impact of the COVID-19 vaccination campaign in Brazil on reported deaths during its first year. Overall, I found the paper well-written and addressing a relevant topic. However, I have a few suggestions that could improve the clarity of the manuscript. Major comments Data Aggregation: The paper does not provide a clear description of how the data were aggregated. While the authors mention using a 15-day moving average to reduce variability in the notification process, it is not clear at what level data were aggregated (age-group X time or age-groupXmunicipalityXtime). Providing more details about the structure of data used for the analyses would be helpful. Treatment: The manuscript does not make it clear what the treatment variable is for the impact evaluation using difference-in-differences (DiD) analysis. While the authors state that each age group was considered the treatment group, and the start of the vaccination rollout was the intervention onset, it is not clear whether the intervention onset is the date the campaign was launched nationally (January 17, 2021) or some other unspecified date. Additionally, if there is variation in timing for different age groups in different municipalities, it is unclear why the authors do not use that variation. Control group: The authors mention that the control group is represented by the national age-adjusted mortality rates in Brazil. If that is the case, the control group would be composed of the treatment groups by definition, leading to spillover effects. The use of rate ratios instead of age-specific death rates leads to similar spillover problems. Clarifying this point would be helpful because this would severely threaten the validity of the design. Formalization: A formal presentation of the DiD model would be helpful to discuss the issues mentioned above. It could be an
--

	equation, like the one presented for the linear regression model used to predict mortality rates for age groups (equation 1). Assumptions: The DiD analysis relies on the parallel trends assumption. Given that different age groups were affected differently by the pandemic, and the evolution in time of outcome variables (e.g., mortality rates) was not necessarily linear, this seems like a strong assumption. Providing the evolution of age-specific death rates before the vaccination campaign could help support this assumption. Minor comments In the Introduction, citing some references about the existing evidence on prioritization in vaccination campaigns would be helpful. Could the authors mention why they start considering outcome variables over a month (February 26) after the campaign started (January 17). Please, Clarify the subscript i in equation (1). Would it be possible to use 2021 population estimates instead of 2020 estimates? When referencing figures and tables in the appendix, please mention the table/figure number in addition to the page number. Discussing how vaccinating high-risk individuals based on comorbidities without following the age-based calendar might affect the results could enrich the manuscript.
--	--

REVIEWER	Nunes, Marta Center for Excellence in Respiratory Pathogens
REVIEW RETURNED	18-Apr-2023

GENERAL COMMENTS	The authors correlated the first year of COVID-19 vaccination rollout in Brazil with the number of COVID-related reported deaths in different age groups. They did this by analysing the temporal association of vaccination coverage, with COVID-19 reported deaths and modelled the prevented and possible preventable deaths. While they try to answer an important question, i.e. if the adopted aged-based vaccination strategy prevented deaths across all age-groups equally and used an interesting analytical approach I have a few comments and suggestions. General comments: The term “preventable deaths” should be introduced as “possible preventable deaths”. Would be useful to have a general description of the NPIs in place during the study period. Can the authors comment or hypothesize on how regional-specific analyses would look in such a large country as Brazil? The problems posed by ecological studies should be better stressed, in particular, that no individual level data was used and that the authors didn’t control for any co-founder effects. The written English needs to be properly reviewed. Specific comments: Conclusion in the abstract: not really correct that “did not significantly prevent the deaths of the youngest”, it was just delayed compared to the older age group. Please re-phrase. Page 6: not clear at this % is refereeing to “with a confirmed case in Brazil comprising 96% of the population.” In the same
--

	paragraph mentioning of “hospital and ICU admissions” is repeated. Page 7, line 58: unclear what mortality was used for the control group “The control group is represented by the national age-adjusted mortality rates in Brazil”. Are these age-adjusted rates for COVID-related deaths (or non-COVID related deaths or overall deaths), occurring during the same period (or in what period)? Throughout the manuscript the authors need to carefully mention that they are referring to COVID-related deaths. In the discussion the authors should mention that one limitation was also that they didn’t account for previous infection. Since the elderly had the highest rate of infection early in the pandemic the natural immunity in this group might have also contributed to some of the protection observed against death. Table 1, please clarify if for the reported deaths, the baseline estimates are the sum of the 6 months or the monthly average? It would be useful to have both estimates. Table 2, the calls in footnote are not on the table. Figure 3, shouldn’t the black dashed line (period when 10% of first dose vaccination coverage was achieved) be placed in different at different times for each age group? Now it is beginning of Jan for all age groups.
--	--

VERSION 1 – AUTHOR RESPONSE

REVIEWER COMMENTS 1

Thank you for the opportunity to review this paper on the impact of the COVID-19 vaccination campaign in Brazil on reported deaths during its first year. Overall, I found the paper well-written and addressing a relevant topic. However, I have a few suggestions that could improve the clarity of the manuscript.

Dear reviewer,

Thank you for this feedback. We are glad to know that our manuscript is well-written and addressing a relevant topic. We are grateful to have the opportunity to review and improve our paper.

Major comments

1. **RW 1 – Data Aggregation:** The paper does not provide a clear description of how the data were aggregated. While the authors mention using a 15-day moving average to reduce variability in the notification process, it is not clear at what level data were aggregated (age-group X time or age-groupXmunicipalityXtime). Providing more details about the structure of data used for the analyses would be helpful.

AUTHORS: Thank you very much for this comment. We do not use individualized data in this study, but daily aggregated data of both COVID-19 reported deaths and vaccine doses. The data was stratified by age group and no further stratification was carried out, except to perform the DID analysis, when the data were stratified by month. To provide a better understanding, we complement the data information as follow:

“Our primary outcome was the estimated number of prevented COVID-19 deaths. Secondary outcomes comprise possible preventable deaths, the potential years of life lost (PYLL), the mortality rates per 100,000 population, and the first dose and second or single dose vaccination coverage. All quantities, rates, and estimates were evaluated temporally for the whole country and stratified by age groups; no further stratification was performed.”

2. **RW 1 – Treatment:** The manuscript does not make it clear what the treatment variable is for the impact evaluation using difference-in-differences (DiD) analysis. While the authors state that each age group was considered the treatment group, and the start of the vaccination rollout was the intervention onset, it is not clear whether the intervention onset is the date the campaign was launched nationally (January 17, 2021) or some other unspecified date. Additionally, if there is variation in timing for different age groups in different municipalities, it is unclear why the authors do not use that variation.

AUTHORS: Thank you for calling our attention to this matter. In the DiD analysis, the intervention onset was the vaccination roll out start nationally (January 17, 2021). With respect to the variation in timing at municipality level, information about the vaccination strategy execution for each city was not available for this analysis. We included more details in the manuscript as follows:

First, we estimated the dynamic effects of the vaccination campaign using a differences-in-differences (DID) analysis.¹⁷ DID is a causal model approach used for impact evaluation by comparing treatment and control groups, before and after an intervention, under non-experimental settings (e.g., non-randomized data).¹⁸ We considered reported COVID-19 deaths as the response variable. Each age group was considered the "treatment variable" and the start of the vaccination campaign roll-out (January 17, 2021) was the intervention onset indicator for all groups. The control group is represented by the national age-adjusted mortality COVID-19 rates in Brazil as a surrogate of the pandemic progression in the country during the vaccination campaign. We compared the daily mortality rates of each age group and the control group using the Rate Ratio (RR). Due to the lack of a proper control group to perform the analysis, using the national age-adjusted mortality could be considered as an alternative to a synthetic control to overcome this difficulty.

3. **RW 1 – Control group:** The authors mention that the control group is represented by the national age-adjusted mortality rates in Brazil. If that is the case, the control group would be composed of the treatment groups by definition, leading to spillover effects. The use of rate ratios instead of age-specific death rates leads to similar spillover problems. Clarifying this point would be helpful because this would severely threaten the validity of the design.

AUTHORS: Thank you for this valuable comment. We discussed this issue extensively during the analysis plan. We used the control group represented by the national age-adjusted mortality rates in Brazil as we sought to inquire about the dynamics of mortality rates for each age group and not to carry out an analysis of vaccine effectiveness itself. We recognize that the chosen control group have the potential for spillover effect, particularly if it would be implemented in a classic DiD to perform an effectiveness analysis. Nonetheless, having an external comparator, such as a set of locations is difficult once there are no countries with the same geographic profile or even the same vaccination strategy that was used by Brazil. Thus, we used the national average, which has in it part of spill-in from those treated, but can provide a relative measure of intervention compared with the national average. Considering the time and spatial variation of the intervention and the size of Brazil, associated with the dynamic monthly specification of the model, we assume the spillover effect is minimal and, ends up being conservative for the estimation of the intervention.

4. **RW 1 – Formalization:** A formal presentation of the DiD model would be helpful to discuss the issues mentioned above. It could be an equation, like the one presented for the linear regression model used to predict mortality rates for age groups (equation 1).

AUTHORS: Thank you for this important suggestion. We included the equation representation for the Negative Binomial Regression model as depicted:

“Afterwards, we applied a Negative Binomial Regression model to estimate the intervention effect.

The response variable (Y_{it}) was the reported deaths, and the covariates were the intervention (age) / control group (X_{it}), the intervention period indicator (I_{it}), their interaction ($X_{it} \cdot I_{it}$), and the population (N_{it}) as the offset variable. To obtain the dynamic effects for each age group, we split the intervention period variable into separate indicators, one for each month. The

intervention effect was calculated as the exponentiated coefficient of the interaction terms, defined as the ratio of rate ratios (RRR)¹⁹ or the DID estimator, and their corresponding 95% confidence intervals. The Negative Binomial Regression model is represented as follows:

$$(1)$$

Where y_{it} represents the monthly observations and α , β , and γ are regression coefficients.”

- RW 1 – Assumptions:** The DiD analysis relies on the parallel trends assumption. Given that different age groups were affected differently by the pandemic, and the evolution in time of outcome variables (e.g., mortality rates) was not necessarily linear, this seems like a strong assumption. Providing the evolution of age-specific death rates before the vaccination campaign could help support this assumption.

AUTHORS: Thank you for pointing this out to us. We recognize that the parallel trend assumption is a problem that may happen in many studies, potentially resulting on many difference-in-difference estimators to be biased. As one of the central points of our work is the dynamic behavior of the risk of death in age groups over time and not the effectiveness of the vaccine itself, we relaxed the parallel trends assumption. We agree that the evolution in time of the mortality rates was not necessarily linear, but comparing to the control group they have similar pattern. Thus, we generate the evolution of age-specific death rates before the vaccination campaign to support the choice of DiD, despite the assumption of linearity not being strongly present before the vaccination roll-out as depicted in the next figure, which will be included in the supplementary material:

Figure S1. Deaths per 100.000 pop stratified by age group – The vertical dashed line refers to the start of the vaccination campaign for each age group.

Minor comments

- RW 1:** In the Introduction, citing some references about the existing evidence on prioritization in vaccination campaigns would be helpful.

AUTHORS: Thank you for this valuable comment. Following your suggestion, we include some references in the introduction reinforcing the evidence on prioritization in vaccination campaigns as can be observed:

“It is still unclear how to best prioritize groups for mass vaccination during pandemics/outbreaks such as COVID-19, especially in low-and-middle-income countries (LMIC) with a high burden of the disease and a young population¹⁴⁻¹⁶.”

2. **RW 1:** Could the authors mention why they start considering outcome variables over a month (February 26) after the campaign started (January 17).

AUTHORS: Thank you very much for this comment. In this study, February 26 corresponds to the beginning of the COVID-19 pandemic, i.e., in the year of 2020. In turn, January 17, is related to the start of the vaccination campaign, in 2021. To avoid any misunderstanding, we include the year when referring to a date.

3. **RW 1:** Please, Clarify the subscript i in equation (1).

AUTHORS: Thank you for calling our attention on this matter. We include a short explanation of the subscript i in equation (1) as can be seen as follow:

“The model is expressed by:

(2)

Where ϵ_t denotes the error term, y_{it} the daily observation for a specific age group and β_0 and β_1 are regression coefficients.”

4. **RW 1:** Would it be possible to use 2021 population estimates instead of 2020 estimates?

AUTHORS: Thank you for this relevant comment. Since we are evaluating the vaccination rollout period (2021), we opted to use population estimates from a baseline period (2020) for rates. As the values are projections, if we use estimates for the period of 2021, this may lead to increased bias, as, for instance, population has been impacted from the deaths during the pandemic. Hence, we opted for a conservative approach of using 2020 population estimates.

5. **RW 1:** When referencing figures and tables in the appendix, please mention the table/figure number in addition to the page number.

AUTHORS: Thank you for calling our attention on this matter. According to the journal format we incorporated the table and figure number can be seen as follow:

*“After September 2021, rate ratios for all age groups returned to similar levels to those seen before vaccination (**online supplemental table S3**).*

*We estimate the intervention effect using a Negative Binomial Regression model to evaluate the association between the reported deaths, the intervention (age)/ control group, and the intervention period indicator (**figure 2-B**). When analysing the temporal progression of the RRR stratified by age group (**online supplemental table S4**), the 70+ age group had the lowest rates in June 2021 (RRR [95% CI]: 0.48 [0.43-0.53]), which evidenced a 52% decrease in the number of deaths in 5 months.”*

6. **RW 1:** Discussing how vaccinating high-risk individuals based on comorbidities without following the age-based calendar might affect the results could enrich the manuscript.

AUTHORS: Thank you very much for this suggestion. The prioritization of high-risk groups during the vaccination roll-out may imply in early protection of these population, especially those with comorbidities. However, our vaccination dose data indicate that, during the first year of the pandemic, only 10% of doses were administered in the comorbidity group, in which average age is around 40 years. Hence, this low proportion may not significantly impact the overall progression of the mortality rates in all ages. We include, in the discussion section, a short paragraph about his matter as can be seen as follow:

“We point out the analysis did not discriminate between prioritized high-risk groups from those individuals that following the age-based calendar in the vaccination campaign, since this is a low percentage of doses from these groups, especially those based on comorbidities lower than 10%, hence not significantly impacting the results or the progression of the vaccination campaign.”

REVIEWER COMMENTS 2

The authors correlated the first year of COVID-19 vaccination rollout in Brazil with the number of COVID-related reported deaths in different age groups. They did this by analyzing the temporal association of vaccination coverage, with COVID-19 reported deaths and modelled the prevented and possible preventable deaths. While they try to answer an important question, i.e. if the adopted aged-based vaccination strategy prevented deaths across all age-groups equally and used an interesting analytical approach I have a few comments and suggestions.

Dear reviewer,

We are glad to read your comments. Thank you very much for giving us the opportunity to organize and implement your considerations in order to improve the quality of the article to fit the standards required by the journal.

General comments

1. **RW 2:** The term “preventable deaths” should be introduced as “possible preventable deaths”.

AUTHORS: Thank you very much for this valuable comment. In order to meet this consideration, we are pleased to inform we re-write this term along the document, according to your suggestion.

2. **RW 2:** Would be useful to have a general description of the NPIs in place during the study period.

AUTHORS: Thank you for pointing this out to us. In order to meet this requirement, we included the following paragraph in the discussion section as follows:

“Although non-pharmacological interventions (NPIs) such as social distancing and masks mandates have been implemented in Brazil, there was not a coordinated and unified guideline at the national level. Due to the country heterogeneity, municipalities applied NPIs under different flags, but not with a high degree of adherence; despite this when the vaccination campaign began none NPIs continued.”

3. **RW 2:** Can the authors comment or hypothesize on how regional-specific analyses would look in such a large country as Brazil?

AUTHORS: Thanks for this interesting question. The vaccination roll-out started at the same time across the country at a slow pace, due to the vaccine’s availability. However, because of Brazil’s continental proportions, regional differences are significant and impose a layer of complexity on this analysis, as COVID-19 spread in different ways and at different speeds. According to Bastos et al., 2022, regional differences can indeed impact the speed of achieving certain vaccine coverage, due to access or the age composition of certain locations. We include this aspect, in the discussion section, as a limitation as follows:

“Third, although the vaccination roll-out started at the same time across the country, regional differences can indeed impact the vaccine coverage speed due to limited access or the age composition of certain locations.¹³ Fourth, we did not take into account for previous infection, once the elderly had the highest rate of infection early in the pandemic, the natural immunity in this group might have also contributed to the same protection observed against death.”

4. **RW 2:** The problems posed by ecological studies should be better stressed, in particular, that no individual level data was used and that the authors didn’t control for any co-founder effects.

AUTHORS: Thank you very for this comment. Using aggregated data, we could adjust for age, very likely the most important confounder in terms of severe COVID-19. The reviewer is correct, so we discussed in the revised version that we did not control for comorbidities, which are likely to have worse immune response to the vaccines and increased risk of severe COVID-19.

5. **RW 2:** The written English needs to be properly reviewed.

AUTHORS: Thank you for calling our attention on this matter. We share your concern to the reading comprehension, and to meet your consideration, the manuscript was edited by a professional native English Language Trainer to eliminate possible grammatical or spelling errors and to conform to accurate scientific English. We would like to reinforce that we have systematically checked and incorporated each all recommendations you made to guarantee accurate and quality of the manuscript.

Specific comments

1. **RW 2:** Conclusion in the abstract: not really correct that “did not significantly prevent the deaths of the youngest”, it was just delayed compared to the older age group. Please re-phrase.

AUTHORS: Thank you for this feedback. We agree and incorporate the following adjustments into the paragraph:

“The adopted aged-based calendar vaccination strategy initially reduced mortality in the oldest but did not prevent deaths of the youngest as effectively as compared to the older age group. Countries with a high burden, limited vaccine supply and young populations should consider other factors beyond age to prioritize who should be vaccinated first.”

2. **RW 2:** Page 6: not clear at this % is refereeing to “with a confirmed case in Brazil comprising 96% of the population.” In the same paragraph mentioning of “hospital and ICU admissions” is repeated.

AUTHORS: Thank you for this comment. The SIVEP-Gripe as nationwide surveillance open-access database used to monitor severe acute respiratory infections (SARI) and comprise the COVID-19 in-hospital reported cases in the country. To a better understanding we adjusted this paragraph a eliminate the repeated words, as can be seen in the following paragraph:

“SIVEP-Gripe has been the primary source of reported COVID-19 deaths in the country. This database covers all Brazilian municipalities; however, in-hospital people belong to 80% of these municipalities, where those territories correspond to 96% of Brazil’s population.² Each register contains information about the individual’s demographics, self-reported symptoms, comorbidities, hospital and ICU admission and ventilatory support, in-hospital outcome (death or discharge), and dates of symptom onset, hospital admission, and ICU admission.”

3. **RW 2:** Page 7, line 58: unclear what mortality was used for the control group “The control group is represented by the national age-adjusted mortality rates in Brazil”. Are these age-adjusted rates for COVID-related deaths (or non-COVID related deaths or overall deaths), occurring during the same period (or in what period)? Throughout the manuscript the authors need to carefully mention that they are referring to COVID-related deaths.

AUTHORS: Thank you for calling our attention on this matter. The used age-adjusted mortality rates correspond to COVID-19 death that occurred six months before vaccination roll-out. In this case, we clarified this information by adding these specifications, as follow:

“We considered reported COVID-19 deaths as the response variable. Each age group was considered the “treatment variable” and the start of the vaccination campaign roll-out (January 17, 2021) was the intervention onset indicator for all groups. The control group is represented by the national age-adjusted mortality COVID-19 rates in Brazil as a surrogate of the pandemic progression in the country during the vaccination campaign.”

4. **RW 2:** In the discussion the authors should mention that one limitation was also that they didn't account for previous infection. Since the elderly had the highest rate of infection early in the pandemic the natural immunity in this group might have also contributed to some of the protection observed against death.

AUTHORS: Thank you very much for this valuable contribution. We are pleased to inform we included the suggested limitation in the discussion section, as can be seen as follow:

“Third, although the vaccination roll-out started at the same time across the country, regional differences can indeed impact the vaccine coverage speed due to limited access or the age composition of certain locations.¹³ Fourth, we did not take into account for previous infection, once the elderly had the highest rate of infection early in the pandemic, the natural immunity in this group might have also contributed to the same protection observed against death. Furthermore, the study has strengths: this is a nationwide data analysis of a large LMIC with a high burden of COVID-19.”

5. **RW 2:** Table 1, please clarify if for the reported deaths, the baseline estimates are the sum of the 6 months or the monthly average? It would be useful to have both estimates.

AUTHORS: Thank you for your suggestion. The baseline values, related to the reported deaths, are the sum of the 6 previously months. We share your concern regard to the tables updating and we included another column for the corresponding to the monthly average. In order to meet the journal guidelines, it was necessary to move this table from the main document to the Supplemental Material file. The table is shown as follow:

1

Table S3. Number of reported deaths and first dose vaccine coverage per month in during the COVID-19 vaccination campaign in Brazil (started in January 17, 2021).

Age group	Monthly Average Baseline*	Baseline*	Jan/21	Feb/21	Mar/21	Apr/21	May/21	Jun/21	Jul/21	Aug/21	Sep/21	Oct/21	Nov/21	Dec/21
Monthly reported deaths														
BR	20,627	123,763	29,346	28,141	62,603	79,939	53,772	48,416	31,293	18,485	12,164	8,254	4,958	3,223
20 – 49	2,050	12,298 (10%)	2,842 (10%)	3,149 (11%)	9,099 (15%)	13,299 (17%)	10,617 (20%)	12,620 (26%)	8,316 (27%)	3,714 (20%)	1,626 (13%)	832 (10%)	506 (10%)	372 (12%)
50 – 59	2,545	15,269 (12%)	3,641 (12%)	3,748 (13%)	9,611 (15%)	14,224 (18%)	11,623 (22%)	13,082 (27%)	7,677 (25%)	3,022 (16%)	1,449 (12%)	830 (10%)	531 (11%)	374 (12%)
60 – 69	4,731	28,385 (23%)	6,845 (23%)	6,588 (23%)	15,385	21,775	15,319	9,363 (19%)	5,210 (17%)	3,471 (19%)	2,276 (19%)	1,634 (20%)	1,063 (21%)	770 (24%)

					(25 %)	(27 %)	(28 %)								
70 +	11,302	67,8 11 (55 %)	16, 01 8 (55 %)	14, 65 6 (52 %)	28, 50 8 (46 %)	30, 64 1 (38 %)	16, 21 3 (30 %)	13, 35 1 (28 %)	10, 09 0 (32 %)	8,2 78 (45 %)	6,8 13 (56 %)	4,9 58 (60 %)	2,8 58 (58 %)	1,7 07 (53 %)	
Monthly average mortality rates per 100,000 people															
BR		0.44	0.6 2	0.6 6	1.3 3	1.7 6	1.1 4	1.0 6	0.6 7	0.3 9	0.2 7	0.1 8	0.1 1	0.0 7	
20 – 49		0.07	0.0 9	0.1 2	0.3 0	0.4 5	0.3 5	0.4 3	0.2 7	0.1 2	0.0 6	0.0 3	0.0 2	0.0 1	
50 – 59		0.35	0.4 9	0.5 6	1.3 0	1.9 9	1.5 7	1.8 3	1.0 4	0.4 1	0.2 0	0.1 1	0.0 7	0.0 5	
60 – 69		0.92	1.3 2	1.4 1	2.9 7	4.3 4	2.9 5	1.8 7	1.0 0	0.6 7	0.4 5	0.3 2	0.2 1	0.1 5	
70 +		2.73	3.8 4	3.8 9	6.8 3	7.5 9	3.8 8	3.3 1	2.4 2	1.9 8	1.6 9	1.1 9	0.7 1	0.4 1	
Vaccine coverage# (%)															
BR			1.8 2	4.7 4	13. 75	21. 92	32. 22	50. 82	67. 72	84. 82	88. 32	89. 58	90. 3	90. 67	
20 – 49			1.9 9	3.6 1	4.5 6	5.9 8	12. 55	28. 85	52. 75	78. 55	83. 55	85. 26	86. 23	86. 72	
50 – 59			1.6 5	3.0 6	4.0 8	7.6 4	31. 84	81. 34	89. 73	92. 36	93. 57	94. 16	94. 53	94. 75	
60 – 69			1.0 5	2.2 8	19. 58	75. 48	95. 28	97. 61	98. 79	99. 6	10 9	10 0.4	100 .6	0.7 2	
70 +			1.8 5	18. 95	90. 35	96. 32	97. 4	97. 99	98. 38	98. 72	99. 07	99. 4	99. 53	99. 61	

*Baseline period corresponds to the six-month period (July to December 2020), previous to the vaccination campaign start. For monthly reported deaths baseline is the sum of the six months deaths.

#At least one dose.

1

6. **RW 2:** Table 2, the calls in footnote are not on the table.

AUTHORS: Thank you for calling our attention on this matter. With the intention of addressing the consideration stated we included both footnotes on the corresponding place, as can be seen as follow:

Table 1. Comparison between observed and predicted deaths during the vaccination campaign in Brazil, in 2021, stratified by age group.

Results	Age group				Total
	20-49	50-59	60-69	70+	
Observed deaths	65,582	67,991	86,126	145,812	365,511
Expected deaths	39,393	48,098	85,064	197,777	370,332
Prevented deaths	155	343	6,032	53,088	59,618
Possible preventable deaths	26,344	20,236	7,094	1,123	54,797
Prevented PYLL †#	6,355	7,203	66,352	291,984	371,894
Preventable PYLL †#	1,080,104	424,956	78,034	6,177	1,589,271

Prevented deaths were obtained if observed deaths was lower than predicted, and possible preventable deaths, if the number of observed deaths was higher than predicted.

†The denominator is the adult population of Brazil for this age group = 151,778,738.

PYLL was calculated using the average age for each group and a life expectancy of 76 years for Brazil in 2020.

7. **RW 2:** Figure 3, shouldn't the black dashed line (period when 10% of first dose vaccination coverage was achieved) be placed in different at different times for each age group? Now it is beginning of Jan for all age groups.

AUTHORS: Thank you for pointing this out to us. We carefully reviewed this figure and perceived that we removed the lines corresponding to 10% of first dose vaccination coverage (placed different at different times), because the graphics were too polluted. In this case, we only considered for in all the age groups, the start of the vaccination roll-out, January 17, 2021. Therefore, we adjusted the legend as follow:

“Figure 3. *Estimated prevented and possible preventable deaths stratified by age group. For each age group (coloured lines), the predicted deaths (black lines) were obtained whether that age-group followed the national age-adjusted mortality rates. Grey lines correspond to the fitted linear model. Predicted deaths were estimated from the period when 10% of first dose vaccination coverage was achieved for each age group. Prevented deaths (green-shaded area) were obtained if observed deaths was lower than predicted, and possible preventable deaths (red-shaded area), if the number of observed deaths was higher than predicted. The vertical dashed lines refer to the start of the vaccination campaign on January 17, 2021.”*

1

VERSION 2 – REVIEW

REVIEWER	Mrejen, Matías Institute for Health Policy Studies
REVIEW RETURNED	19-Sep-2023
GENERAL COMMENTS	Comments for editor and author Thank you for the opportunity to review the revised version of this paper on the impact of the COVID-19 vaccination campaign in Brazil on reported deaths during its first year. I have some relevant

	remaining concerns regarding the author's methodological approach that I believe need to be addressed: Control group: The authors constructed a control group using an average of the outcomes for different treatment groups, which inherently violates the causal assumption behind difference-in-differences analysis. This assumption posits that the evolution of outcomes in the control group provides a valid counterfactual for what the trend in the treatment group would have been if they had not received the intervention. In this case, the control group has effectively received the intervention, rendering the assumption invalid by definition. Treatment: The decision to treat all age groups as if they began receiving the intervention at the outset of the vaccination campaign roll-out also presents issues. It results in the absence of a proper control group. A more effective approach would be to obtain more granular, age-specific onset dates for the vaccination campaign roll-out. This would enable the implementation of a staggered difference-in-differences analysis, where, in each period, the not-yet-treated age groups would serve as the control group for the treated groups. Conclusion and recommendations: In light of the aforementioned concerns, I believe the authors' current methodology does not allow them to achieve the objective stated in the introduction, which is to "evaluate the impact of the vaccination campaign in Brazil on the progression of reported COVID-19 deaths." However, I believe that an alternative analysis that examines how age-specific death rates deviated from the national average before and after the onset of the vaccination campaign could still provide valuable insights. This approach would necessitate a revision of the methodology and the writing to reflect the study's limitations more accurately.
--	--

REVIEWER	Nunes, Marta Center for Excellence in Respiratory Pathogens
REVIEW RETURNED	01-Sep-2023

GENERAL COMMENTS	my comments have been properly addressed.
---

VERSION 2 – AUTHOR RESPONSE

REVIEWER COMMENTS 1

Thank you for the opportunity to review the revised version of this paper on the impact of the COVID-19 vaccination campaign in Brazil on reported deaths during its first year.

Dear reviewer,

Thank you very much for giving us the opportunity to organize and implement your considerations in order to improve the quality of the article to fit the standards required by the journal.

General comments

I have some relevant remaining concerns regarding the author's methodological approach that I believe need to be addressed:

RW – Control group: The authors constructed a control group using an average of the outcomes for different treatment groups, which inherently violates the causal assumption behind difference-in-differences analysis. This assumption posits that the evolution of outcomes in the control group provides a valid counterfactual for what the trend in the treatment group would have been if they had not received the intervention. In this case, the control group has effectively received the intervention, rendering the assumption invalid by definition.

RW – Treatment: The decision to treat all age groups as if they began receiving the intervention at the outset of the vaccination campaign roll-out also presents issues. It results in the absence of a proper control group. A more effective approach would be to obtain more granular, age-specific onset dates for the vaccination campaign roll-out. This would enable the implementation of a staggered difference-in-differences analysis, where, in each period, the not-yet-treated age groups would serve as the control group for the treated groups.

RW – Conclusion and recommendations: In light of the aforementioned concerns, I believe the authors' current methodology does not allow them to achieve the objective stated in the introduction, which is to "evaluate the impact of the vaccination campaign in Brazil on the progression of reported COVID-19 deaths." However, I believe that an alternative analysis that examines how age-specific death rates deviated from the national average before and after the onset of the vaccination campaign could still provide valuable insights. This approach would necessitate a revision of the methodology and the writing to reflect the study's limitations more accurately.

AUTHORS RESPONSE: Thank you for calling our attention to this matter. We agree with you that the constructed "control group" using an average of the outcomes for different treatment groups, i.e., the national age-adjusted mortality COVID-19 rates in Brazil, violates the causal assumption of difference-in-differences analysis. Additionally, in the treatment case, due to the difficulty to obtain more granular information about the vaccination strategy execution for each city and age group, it will not be possible to tackle this aspect as you indicate. Therefore, we will adapt the suggestions made in the "**Conclusion and recommendations**" with the aim of providing close evidence to the initial results, but which methodologically satisfy the premises of the implemented modelling.

Following your recommendation, and encouraged by the Difference-in-Difference analysis, we examine how age-specific death rates deviated from the national average before and after the onset of the vaccination campaign, we adjusted the manuscript as follows:

"First, we estimated the dynamic effects of the vaccination campaign to examine how age-specific death rates deviated from the national average before and after the onset of the vaccination campaign. The national average is represented by the national age-adjusted mortality COVID-19 rates in Brazil as a surrogate of the pandemic progression in the country during the vaccination campaign. We compared the daily mortality rates of each age group and the national age-adjusted mortality COVID-19 rates using Rate Ratios (RR). Afterwards, we applied a Negative Binomial Regression model. The response variable () was the monthly COVID-19 reported deaths, and the covariates were the age group (), the period indicator (), separated into indicators, one

for each month, their interaction (), and the population () as the offset variable. To obtain the dynamic effects for each age group, we used the national average as the reference. The effect was calculated as the exponentiated coefficient of the interaction terms, defined as the Rate Ratio (RR) and their corresponding 95% confidence intervals. The Negative Binomial Regression model is represented as follows

$$(1)$$

where represents the monthly observations and , , and are regression coefficients. Additionally, a sensitivity analysis is performed by exchanging the national average as reference to the older and younger (70+ and 20-49) age groups.”

“...We estimated the dynamic effect of age-specific death rates and how they deviated from the national average using a Negative Binomial Regression model to evaluate the association between the reported deaths, age group and the period indicator (**figure 2-B**). When analysing the temporal progression of the RR stratified by age group (**online supplemental table S5**), the 70+ age group had the lowest rates in June 2021 (RR [95% CI]: 0.48 [0.43-0.53]), which evidenced a 52% decrease in the number of deaths in 5 months. With regard to age groups 50-59 and 60-69, we estimated a decrease in death rates of 15% and 30% (RR [95% CI]: 0.85 [0.75-0.96] and RR [95% CI]: 0.70 [0.63-0.71], respectively) for at least 4 and 6 months in the first year of vaccination (**online supplemental table S5**).

When performing the sensitive analysis (**online supplemental figure S3 and tables S6-S7**), the extreme age groups (20-49 and 70+) display a similar behaviour from those age groups obtained from the dynamic effect when comparing to the national average (**figure 2-B**). Furthermore, evaluating the before and after the vaccination campaign (reference Dec/20) was evidenced a rise on the mortality rates (**online supplemental figure S1-B and table S8**) after the start of the vaccination campaign which became increasingly higher for younger age groups (20-49 and 50-59). In June there was a second increase that was not repeated for the older age groups (60-69 and 70+).

Figure S1. (A) Deaths per 100.000 pop stratified by age group – The vertical dashed line refers to the start of the vaccination campaign. (B) Estimated effect for each age group before and after the beginning the vaccination campaign. Effects were obtained as the Rate Ratio (RR) and their respective 95% confidence intervals.

Figure S3. (A) Estimated effect of the vaccination campaign in each age groups comparing to 70+ age group. (B) Estimated effect of the vaccination campaign in each age groups comparing to 20-49 age group. Effects were obtained as the Rate Ratio (RR) and their respective 95% confidence intervals. The vertical dashed line refers to the start of the vaccination campaign on January 17, 2021. Our analysis considered six months before the vaccination roll-out as the baseline period and used the national mortality rates as reference in a Negative Binomial Regression model.

In figure S3-A is possible to observe how the 20-49 age group has the highest rates when compared with the 50-59 and 60-69, considering 70+ age group, reaching in Jun/21 an increase of 500% (RR [95% CI]: 5.71 [5.15-6.34]). It is worth mentioning that for the age group 60-69, we estimated a decrease in death rates of 20%, 21% and 13% (RR [95% CI]: 0.80 [0.72-0.89], RR [95% CI]: 0.79 [0.71-0.88] and RR [95% CI]: 0.87 [0.76-1.00], respectively) for the 9th, 10th, and 11th months in the first year of vaccination. The 50-59 age group had a similar behavior with a decline of 21% and 11% (RR [95% CI]: 0.79 [0.71-0.88] and RR [95% CI]: 0.89 [0.79-1.00], respectively) only in the 10th and 11th month (**online supplemental table S6**). With regard to Figure S2-B, considering 20-49 as the age reference group, most of the deaths had a reduction in death rates, attaining the lowest rates for the 70+ age group in Jun/21 with a decrease of 82% (RR [95% CI]: 0.19 [0.16-0.19]) (**online supplemental table S7**).

Table S6. Monthly estimated effect of the vaccination campaign with their respective 95% confidence intervals stratified by age group (70+ reference).

Age group	Jan/21	Feb/21	Mar/21	Apr/21	Mai/21	Jun/21	Jul/21	Aug/21	Sep/21	Oct/21	Nov/21	Dec/21
20 – 49	1.07 (0.96 - 1.19)	1.30 (1.16 - 1.45)	1.93 (1.74 - 2.14)	2.62 (2.37 - 2.91)	3.96 (3.57 - 4.39)	5.71 (5.15 - 6.34)	4.98 (4.49 - 5.53)	2.71 (2.44 - 3.02)	1.44 (1.29 - 1.62)	1.01 (0.90 - 1.15)	1.07 (0.93 - 1.23)	1.32 (1.13 - 1.53)
50 – 59	1.07 (0.96 - 1.19)	1.20 (1.08 - 1.34)	1.58 (1.43 - 1.75)	2.18 (1.97 - 2.41)	3.36 (3.05 - 3.72)	4.60 (4.16 - 5.09)	3.57 (3.23 - 3.96)	1.71 (1.54 - 1.91)	1.00 (0.89 - 1.12)	0.79 (0.70 - 0.89)	0.87 (0.76 - 1.00)	1.03 (0.89 - 1.19)
60 – 69	1.02 (0.93 - 1.13)	1.08 (0.97 - 1.19)	1.29 (1.17 - 1.42)	1.70 (1.54 - 1.87)	2.26 (2.05 - 2.49)	1.68 (1.52 - 1.85)	1.24 (1.18 - 1.37)	1.00 (0.91 - 1.11)	0.80 (0.72 - 0.89)	0.79 (0.71 - 0.88)	0.89 (0.79 - 1.00)	1.08 (0.95 - 1.23)
70+ (REF)	-	-	-	-	-	-	-	-	-	-	-	-

Table S7. Monthly estimated effect of the vaccination campaign with their respective 95% confidence intervals stratified by age group (20-49 reference).

Age group	Jan/21	Feb/21	Mar/21	Apr/21	Mai/21	Jun/21	Jul/21	Aug/21	Sep/21	Oct/21	Nov/21	Dec/21
20 – 49	-	-	-	-	-	-	-	-	-	-	-	-

(RE F)												
50 – 59	0.99 (0.89 - 1.19)	0.92 (0.82 - 1.04)	0.82 (0.74 - 0.92)	0.83 (0.75 - 0.93)	0.85 (0.76 - 0.95)	0.81 (0.72 - 0.90)	0.71 (0.64 - 0.80)	0.63 (0.56 - 0.71)	0.69 (0.61 - 0.79)	0.77 (0.67 - 0.89)	0.81 (0.69 - 0.96)	0.78 (0.65 - 0.93)
60 – 69	0.95 (0.85 - 1.07)	0.82 (0.74 - 0.93)	0.67 (0.60 - 0.74)	0.65 (0.58 - 0.72)	0.57 (0.51 - 0.63)	0.29 (0.26 - 0.33)	0.24 (0.22 - 0.28)	0.37 (0.33 - 0.41)	0.55 (0.49 - 0.63)	0.78 (0.68 - 0.89)	0.83 (0.72 - 0.96)	0.82 (0.70 - 0.96)
70+	0.93 (0.84 - 1.04)	0.76 (0.69 - 0.86)	0.52 (0.47 - 0.57)	0.38 (0.34 - 0.42)	0.25 (0.23 - 0.28)	0.18 (0.16 - 0.19)	0.20 (0.18 - 0.22)	0.37 (0.33 - 0.41)	0.69 (0.62 - 0.78)	0.99 (0.87 - 1.12)	0.93 (0.81 - 1.07)	0.75 (0.65 - 0.88)

Table S8. Monthly estimated effect of the vaccination campaign with their respective 95% confidence intervals stratified by age group.

Age group	Jan/ 21	Feb/ 21	Mar/ 21	Apr/ 21	Mai/ 21	Jun/ 21	Jul/ 21	Aug/ 21	Sep/ 21	Oct/ 21	Nov/ 21	Dec/ 21
20 – 49	1.36 (1.24 - 1.50)	1.67 (1.52 - 1.84)	4.36 (4.00 - 4.76)	6.59 (6.04 - 7.19)	5.09 (4.67 - 5.56)	6.25 (5.73 - 6.83)	3.99 (3.68 - 4.35)	1.78 (1.63 - 1.95)	0.81 (0.73 - 0.89)	0.40 (0.36 - 0.44)	0.25 (0.22 - 0.28)	0.18 (0.16 - 0.20)
50 – 59	1.35 (1.23 - 1.49)	1.55 (1.41 - 1.70)	3.58 (3.28 - 3.91)	5.48 (5.01 - 5.99)	4.33 (3.97 - 4.73)	5.04 (4.61 - 5.51)	2.86 (2.62 - 3.13)	1.13 (1.03 - 1.24)	0.56 (0.50 - 0.62)	0.31 (0.28 - 0.35)	0.20 (0.18 - 0.23)	0.14 (0.12 - 0.15)
60 – 69	1.29 (1.20 - 1.41)	1.38 (1.27 - 1.50)	2.92 (2.70 - 3.15)	4.27 (3.96 - 4.61)	2.91 (2.69 - 3.14)	1.84 (1.70 - 1.99)	0.99 (0.91 - 1.07)	0.66 (0.61 - 0.72)	0.45 (0.41 - 0.49)	0.31 (0.28 - 0.34)	0.21 (0.19 - 0.23)	0.15 (0.13 - 0.16)
70+	1.28	1.29	2.26	2.51	1.29	1.09	0.80	0.66	0.56	0.39	0.23	0.14

	(1.18 - 1.37)	(1.19 - 1.39)	(2.09 - 2.44)	(2.32 - 2.72)	(1.19 - 1.39)	(1.01 - 1.18)	(0.7 4 - 0.87)	(0.61 - 0.71)	(0.52 - 0.61)	(0.36 - 0.43)	(0.22 - 0.26)	(0.12 - 0.15)
--	---------------------	---------------------	---------------------	---------------------	---------------------	---------------------	--------------------------	---------------------	---------------------	---------------------	---------------------	---------------------

VERSION 3 – REVIEW

REVIEWER	Mrejen, Matías Institute for Health Policy Studies
REVIEW RETURNED	23-Jan-2024

GENERAL COMMENTS	No further comments
---------------------